# Unsupervised Discovery of Visual Attributes using non-Gaussianity

## Abstract

In everyday language, we often describe a face with attributes such as "young male, with glasses, smiling." This paper asks whether such attributes can be discovered automatically from large collections of unlabeled images by building on the classical notion of "projection pursuit". Specifically, we search for linear projections of a StyleGAN2 (Karras et al., 2020) latent space whose one-dimensional distributions are maximally non-Gaussian. Across datasets of faces, churches, cars and animals these non-Gaussian directions frequently correspond to meaningful visual attributes, whereas the original latent coordinates, or their principal components, typically do not. Taken together, the results suggest that concepts like "smiling" or "glasses" correspond to statistical properties of image collections and can be uncovered without supervision by exploiting non-Gaussianity.

## 1 Introduction

Modern deep generative models can generate astonishingly realistic, diverse images and enable controllable image editing for a remarkably wide range of tasks (Karras et al., 2019; 2020; Abdal et al., 2020). One hypothesis for the source of this success is that the latent representation learned by these models is aligned with semantic attributes of the image collection, a result that is sometimes referred to as "disentanglement" (Horan et al., 2021). According to this hypothesis, which we call the semantic latent space hypothesis, it is impossible to generate high quality synthetic face images that are both varied and realistic unless the latent space has explicitly learned to represent attributes such as "glasses" and "smiling".

A naive version of the semantic latent space hypothesis is easily falsified, both for theoretical and experimental reasons. Theoretically, any deep generator can be converted into another one that generates exactly the same images but with a different latent representation (by applying an invertible transformation to the latent codes, and the inverse of that transformation to the generator) (Locatello et al., 2019; Horan et al., 2021). Experimentally, it has been shown that for most successful generative models, moving in the latent space in one of the cardinal directions (i.e changing one number in the latent code while keeping all others fixed) will almost always not correspond to a semantic change of the image Voynov & Babenko (2020); Shen & Zhou (2021).

The shortcomings of the naive semantic latent space hypothesis can be understood in two ways. First, the attributes we usually describe in language, such as "smiling," may not align with the true latent variables of human faces. In this view, there could exist a different set of underlying factors that explain the data more faithfully, but for which we simply lack everyday words. Second, semantic directions may in fact exist within the latent space of modern generative models, but they do not align with the coordinate axes of the latent representation (Härkönen et al., 2020; Abdal et al., 2021; Shen et al., 2020).

In this paper we present evidence that semantically meaningful directions *are* present in the latent representation of StyleGAN2 and can be easily recovered using the classical approach of projection pursuit (Friedman & Tukey, 1974). Specifically, our method is based on the classical insight that interesting projections of the data should be non-Gaussian and we show experimentally that this insight holds for semantic directions in the latent space of StyelGAN. This insight motivates our algorithm in which we search for directions that are as non-Gaussian as possible and our experiments suggest that indeed in StyleGAN2, the most non-Gaussian directions are almost always semantic

(i.e. agree with ground truth labels for human faces). Moreover, among the top 500 directions ranked by non-Gaussianity, roughly 30% exhibit clear and easily interpretable semantic effects.

The contributions of our paper include:

- A straightforward application of a classical tool, projection pursuit/ICA, to a modern setting: StyleGAN2 inversions of real, unlabeled images, requiring no retraining or labels.

- A simple link between non-Gaussianity and semantics: directions with non-Gaussian marginals tend to correspond to coherent visual attributes (contrasting with variance-based axes).

- Evidence across faces, churches, and cars: the discovered directions produce consistent edits and show quantitative alignment with lightweight linear probes.

## 2 RELATED WORK

### 2.1 UNSUPERVISED DISCOVERY OF LATENT DIRECTIONS IN GANS

Unsupervised approaches aim to recover interpretable axes in pretrained GANs without labels, either by analyzing latent distributions or by inspecting generator structure. Härkönen et al. (2020) apply PCA to latent codes or intermediate features and edit along principal components, yielding intuitive controls (pose, color, zoom), yet variance maximization does not guarantee semantic alignment and often leaves factors entangled. In a complementary, training-free route, Shen & Zhou (2021) derive a closed-form factorization (SeFa) of early generator weights to expose eigen-directions that induce broad, often global edits; the method is fast but data-agnostic and sensitive to architecture/layer choice. Beyond closed-form or linear decompositions, Voynov & Babenko (2020) learn multiple directions by training an auxiliary predictor to identify which latent shift produced an observed change, encouraging diverse transformations but optimizing for change predictability rather than latent statistics. We compare to their approach in our experiments.

### 2.2 UNSUPERVISED DISENTANGLEMENT

A number of works have attempted to learn a disentangled representation directly from image observations (e.g. Horan et al. (2021); Reizinger et al. (2022); Heurtebise et al. (2025). This is a much more difficult task then the one we are addressing here and is in fact in many cases impossible without strong assumptions or additional information Locatello et al. (2019). In contrast, we focus on the task of finding linear projections of the latent space in a pre-learned generative model.

### 2.3 SUPERVISED AND WEAKLY-SUPERVISED ATTRIBUTE EDITING

Supervised and weakly-supervised approaches seek user-specified controls by leveraging labels, text, or auxiliary predictors. Shen et al. (2020) learn linear SVM hyperplanes in latent space from labeled samples (or classifier-generated labels); the separating normal becomes an editing direction for attributes such as *smile* or *glasses*. Wu et al. (2020) analyze StyleGAN's per-channel style parameters (StyleSpace) and show that sparse, channel-wise manipulations produce localized, disentangled edits; channels can be selected via simple statistics or lightweight supervision. Abdal et al. (2021) introduce conditional continuous normalizing flows to traverse $\mathcal{W}$ according to target attribute values predicted by auxiliary regressors, enabling multi-attribute control while preserving identity. Complementarily, Patashnik et al. (2021) exploit vision–language pretraining: a CLIP-based loss aligns generated images with text prompts, yielding open-vocabulary edits via either optimization or a learned mapper, at the cost of extra computation and occasional artifacts. In contrast, our method uses no external labels or text and ranks candidate directions purely by a data-driven non-Gaussianity criterion.

## 3 THE IMPORTANCE OF NON-GAUSSIANITY

We adopt the projection–pursuit (Friedman & Tukey, 1974) perspective and define *interesting directions* as those along which the one–dimensional projection becomes maximally non-Gaussian. Con-

cretely, for a unit vector $\boldsymbol{w}$ and centered data $\boldsymbol{x}$, the projection $y = \boldsymbol{w}^\top \boldsymbol{x}$ is informative if it departs markedly from Gaussianity. The motivation comes from the **central limit theorem**: linear mixtures of independent non-Gaussian variables tend to be *more* Gaussian than the variables themselves, so directions that push the projection away from Gaussianity (i.e. increase its non-Gaussianity) reveal the true latent structure.

Figure 1 provides evidence that the non-Gaussianity criterion suggested over 50 years ago, may help find semantic directions in the latent space of StyleGAN2. For each CelebA image, we found a latent code in $\mathcal{W}^+$ of StyleGAN2 and we trained linear classifiers in the latent space for the ground truth attributes given in the dataset (e.g. "heavy makeup", "smiling" etc.). We then computed the histogram of the projection of the full CelebA dataset on these semantic directions as well as random directions in the same latent space. As the figure shows, there is a clear distinction between the semantic directions (which have a very non-Gaussian projection, often bimodal) and the random directions (which are always unimodal and fit very well to a Gaussian distribution).

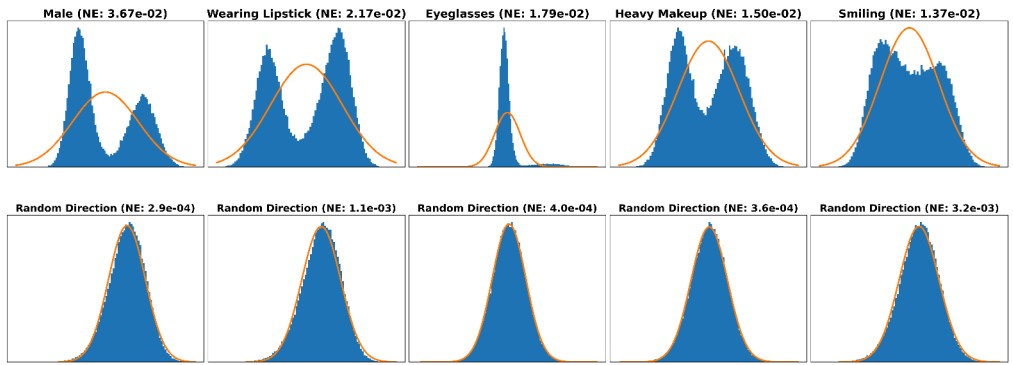

Figure 1: The top row shows histograms of the projections of inverted CelebA samples onto hyperplanes learned from CelebA attributes. The orange curves denote Gaussian distributions with matched mean and variance. The bottom row shows the same samples projected onto random directions in the latent space. While random directions typically yield Gaussian-like histograms, semantically meaningful directions deviate strongly from Gaussianity.

In order to quantify how non-Gaussian a projection is, we use the classical measure of *Negentropy*. For a scalar variable y with the same variance as a Gaussian $\nu$, the *negentropy*

$$J(\mathrm{y}) = H(\nu) - H(\mathrm{y}) \geq 0 \tag{1}$$

quantifies departure from Gaussianity (Hyvärinen, 1999). Here $H$ denotes the *differential entropy*, $H(y) = -\int p_y(t) \log p_y(t) dt$, so that $J(\mathrm{y}) = 0$ iff y is Gaussian and larger values indicate stronger non-Gaussian structure.

The exact differential entropy is difficult to estimate reliably. We follow the approach suggested in the ICA literature Hyvärinen (1999) and maximize a smooth, robust surrogate of negentropy of the form

$$J(y) \approx \big(\mathbb{E}[G(y)] - \mathbb{E}[G(\nu)]\big)^2 \tag{2}$$

where $G$ is a non-quadratic **contrast** (e.g. $G(u) = \log \cosh u$ or $G(u) = -\exp(-u^2/2)$). The figure titles in Figure 1 display $J(y)$ for each histogram. The random directions have a value that is usually two orders of magnitude smaller than the semantic directions. This suggests that indeed $J(y)$ may be a useful numerical method to find semantic directions and our experiments are designed to evaluate this method more quantitaively.

### 3.1 FINDING NON-GAUSSIAN DIRECTIONS

The **FastICA** algorithm (Hyvärinen, 1999) searches for unit vectors $\boldsymbol{w}$ that maximize the negentropy surrogate along $y = \boldsymbol{w}^\top \boldsymbol{x}$. For a broad class of contrasts, this yields a consistent estimator of high-contrast directions. The optimization proceeds via a fixed-point (approximate Newton) iteration

$$\boldsymbol{w} \leftarrow \mathbb{E}\big[\boldsymbol{x} g(\boldsymbol{w}^\top \boldsymbol{x})\big] - \mathbb{E}\big[g'(\boldsymbol{w}^\top \boldsymbol{x})\big]\boldsymbol{w} \tag{3}$$

followed by normalization; when extracting multiple directions, rows of $\boldsymbol{W}$ are orthogonalized at each step. This update avoids step-size tuning and is empirically superlinear.

When the data is generated by a linear mixing of independent sources, then fastICA is guaranteed to find these sources under classical choices of $G$ (e.g., kurtosis $G(u) = u^4/4$) and standard assumptions on the sources up to scale and permutation. However, when the data is not generated by a linear mixing of independent sources, the recovered directions may very well be dependent. In our setting, there is no reason to believe that visual attributes are independent (e.g. "wears lipstick", "female" will tend to correlate in most datasets) and so we are *not* using ICA to find independent directions, rather directions that are as non-Gaussian as possible.

## 4 METHOD

Rather than analyzing synthetic latent codes sampled directly from the prior of StyleGAN2, we ground our study in real-world image data. This ensures that the directions we discover reflect the natural variation present in the world, rather than artifacts of the previous distribution of the generator. We embed real-world datasets into the extended latent space $\mathcal{W}^+$ of StyleGAN2 using the ReStyle encoder (Alaluf et al., 2021). Since not all inversions achieve satisfactory reconstruction quality, we filter out poor reconstructions. By retaining only successful inversions, the resulting latent dataset reliably captures the semantic variability of real images. This step is critical: it ensures that subsequent analysis (e.g., ICA directions) is based on latents that faithfully mirror real-world structure rather than encoding noise or inversion artifacts.

We apply *Independent Component Analysis* (ICA) to the set of latent codes $\{\boldsymbol{w}_i\}_{i=1}^N$. For implementation, we use the `FastICA` algorithm from `scikit-learn` (Pedregosa et al., 2011), configured with the *log-cosh* function as the non-Gaussianity measure.

Concretely, ICA estimates an unmixing matrix $\boldsymbol{A}^{-1}$ such that

$$\boldsymbol{s}_i = \boldsymbol{A}^{-1}\boldsymbol{w}_i,$$

where $\boldsymbol{s}_i$ exhibit non-Gaussian distributions. The corresponding mixing matrix $\boldsymbol{A}$ contains columns $\boldsymbol{d}_j$ that we interpret as candidate latent directions. These directions $\boldsymbol{d}_j$ serve as the basis for all subsequent evaluations, both qualitative and quantitative, where we later compare them with different baselines (see Section 5).

## 5 EXPERIMENTS

### 5.1 EVALUATION PROTOCOL

We evaluated the extent to which the recovered directions are semantic using two complementary approaches. First, we perform qualitative evaluations by interpolating along candidate directions. For a given latent code $\boldsymbol{w}$ and direction $\boldsymbol{d}_j$, we generate

$$\boldsymbol{w}' = \boldsymbol{w} + \alpha\boldsymbol{d}_j, \quad \alpha \in [-\gamma, \gamma],$$

and synthesize the corresponding images. Smooth and interpretable image transformations are treated as evidence of semantic structure.

Second, we conduct a quantitative evaluation using annotated binary attributes available for the CelebA data set. For each attribute (e.g. "smiling", "has lipstick"), we train a linear SVM in the latent space of styleGAN2 which yields a separating hyperplane $(\boldsymbol{w}_a, b)$. The latent space that use throughout the paper is a reduced $\mathcal{W}^+$: it consists of 18 style vectors of dimensions 512 which we reduce to 512 dimensions using PCA. The normal vector $\boldsymbol{w}_a$ has been shown to be semantically meaningful (Shen et al., 2020) and, when trained on diverse data, reasonably disentangled. We therefore treat the set of SVM weight vectors as our *ground-truth semantic directions*. To measure the alignment between a discovered direction $\boldsymbol{d}_j$ and an attribute hyperplane $\boldsymbol{w}_a$, we compute the absolute cosine similarity:

$$\text{align}(\boldsymbol{d}_j, \boldsymbol{w}_a) = \left| \frac{\boldsymbol{d}_j^\top \boldsymbol{w}_a}{\|\boldsymbol{d}_j\|\|\boldsymbol{w}_a\|} \right|$$

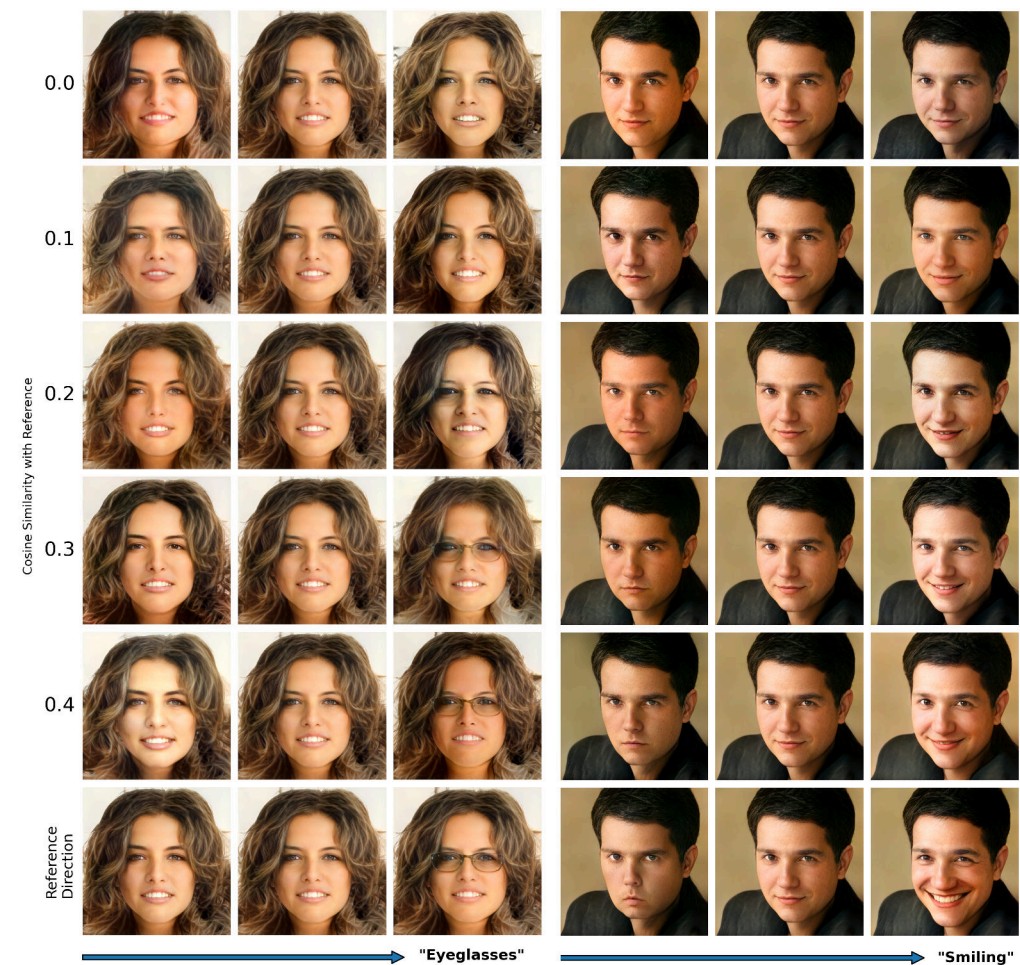

Figure 2: The bottom row shows reference directions for the semantic attributes smiling and glasses. The rows above depict random directions constrained to have varying cosine similarity with these references. As similarity decreases, the semantic effect diminishes. At a similarity of 0.3, residual semantic structure remains visible, supporting the choice of cosine similarity larger than 0.3 as a criterion for successful retrieval of semantic directions.

This ensures that both a direction and its flipped counterpart are treated as equivalent matches, since they induce the same semantic edit up to sign.

We treat direction discovery as a retrieval problem: given $M$ ground-truth attribute directions, the goal is to retrieve them from among $K$ candidate directions. A candidate is considered a successful retrieval if its cosine similarity to some ground-truth direction exceeds $0.3$. This threshold was empirically selected, as values above $0.3$ consistently preserved the semantic meaning of the attribute direction in qualitative interpolations (see Figure 2).

Performance is reported in terms of Precision@k and Recall@k, standard metrics in retrieval tasks (Manning et al., 2008).

## 5.2 BASELINES

We compare ICA-derived directions against three baselines:

- **PCA:** directions derived from performing eigen-decomposition on the covariance matrix of the inverted images' latent codes. The resulting components are ordered by explained variance.

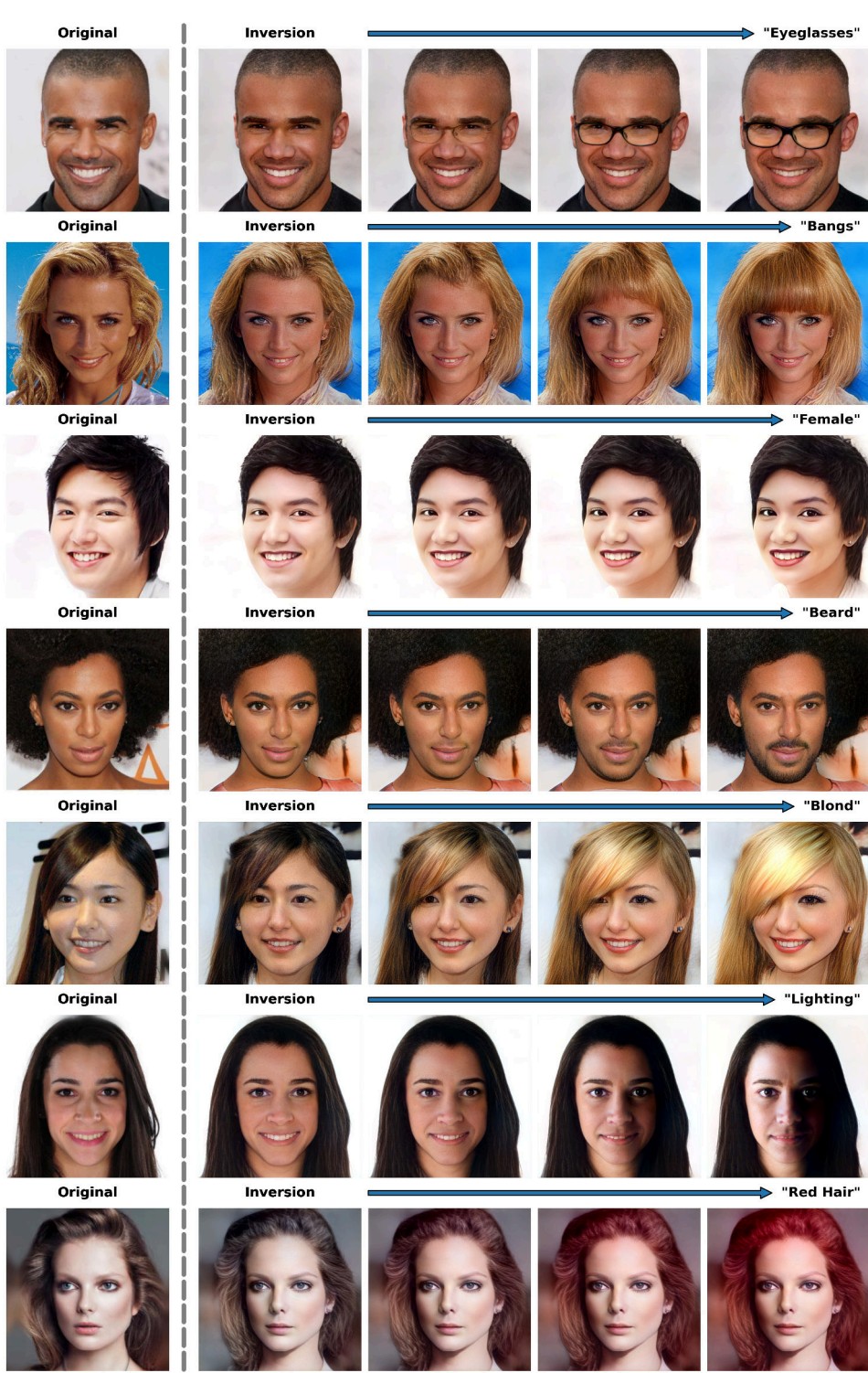

Figure 3: Each row shows a CelebA source image, its inversion, and subsequent interpolations along a semantic direction discovered by ICA. Rows 1–4 correspond to attributes annotated in CelebA, while rows 5–7 illustrate additional semantic directions uncovered by ICA that are not labeled in the dataset.

- **GanLatentDiscovery:** directions produced by the method of Voynov & Babenko (2020), which formulates latent discovery as a supervised classification task. The generator is probed with perturbed codes, and a lightweight auxiliary network is trained to predict the perturbation. The learned classifier weights define a set of editing directions.

- **Random:** directions sampled uniformly from the unit sphere in $\mathbb{R}^{512}$.

- **Standard basis:** coordinate axes in $\mathcal{W}^+$.

## 5.3 DATASETS AND SETUP

We first focus on the CelebA dataset (Liu et al., 2015), which contains approximately 200k celebrity face images. CelebA is a natural choice for three reasons: (1) it reflects a real-world distribution of human faces, (2) it can be reliably inverted into a StyleGAN2 model pretrained on FFHQ (Karras et al., 2019; 2020) using the ReStyle encoder (Alaluf et al., 2021), and (3) it provides 40 binary attribute annotations (e.g., *Eyeglasses*, *No Beard*, *Male*), which serve as supervised signals for quantitative evaluation.

All 200k images were inverted into the extended latent space $\mathcal{W}^+$. We retained 130k high-quality inversions according to a combined MSE and LPIPS(Zhang et al., 2018) reconstruction score. Each inversion in $\mathcal{W}^+$ consists of 18 style vectors of dimension 512. For downstream analysis, we flatten these representations and reduce them to 512 dimensions using PCA. Finally, ICA with log-cosh as the non-Gaussianity proxy is applied, yielding 512 candidate directions corresponding to the columns of the mixing matrix.

Beyond CelebA, we also conducted experiments on three additional datasets: LSUN Church (Yu et al., 2015), Stanford Cars (Krause et al., 2013), and AFHQ Wild (Choi et al., 2020). These datasets broaden the domain beyond human faces to large-scale scene categories (church exteriors), fine-grained object categories (cars), and animal images (wild species). For each dataset we adopted the same inversion pipeline as in the CelebA case: real images were embedded into the extended latent space $\mathcal{W}^+$ of a pretrained StyleGAN2 generator using the ReStyle encoder, followed by filtering to retain only high-quality reconstructions. The resulting latent sets were then reduced to 512 dimensions via PCA and subjected to ICA with the log-cosh non-Gaussianity measure.

Unlike CelebA, however, these datasets do not provide annotated attributes or ground-truth semantic hyperplanes. Consequently, no quantitative retrieval evaluation is possible. Instead, we rely on qualitative analysis: interpolations along the ICA directions are inspected for coherent semantic changes, which we report in the *Qualitative Results* section.

## 5.4 QUANTITATIVE RESULTS

Figure 5 shows precision@k and recall@k for our method compared with four baselines: PCA, random directions, standard basis vectors, and directions obtained by GanLatentDiscovery. For GanLatentDiscovery we tried two versions: we evaluated the directions that were computed by the authors and uploaded to their repository (after projecting them linearly to the latent space used by our method) and we also ran their code directly in the latent space that we use. None of the baselines were able to recover semantic directions (as defined by the cosine similarity with the ground truth binary labels), whereas ICA successfully retrieved nearly 50% of them.

We also examined how the ordering of ICA directions affects retrieval. In one version, directions were randomly shuffled; in the other, they were sorted according to their negentropy. The results clearly show that ordering by negentropy yields better retrieval rates, with high-negentropy directions much more likely to correspond to semantic changes.

Although the precision values seem to be low for all methods (e.g. with K=30 all methods have a precision close to zero) it should be noted that the 40 ground truth annotations are almost surely only a subset of the possible semantic attributes of CelebA. In the next section, we also use a more qualitative evaluation method in order to also evaluate whether ICA found directions that were semantic but did not appear among the 40 ground truth annotations.

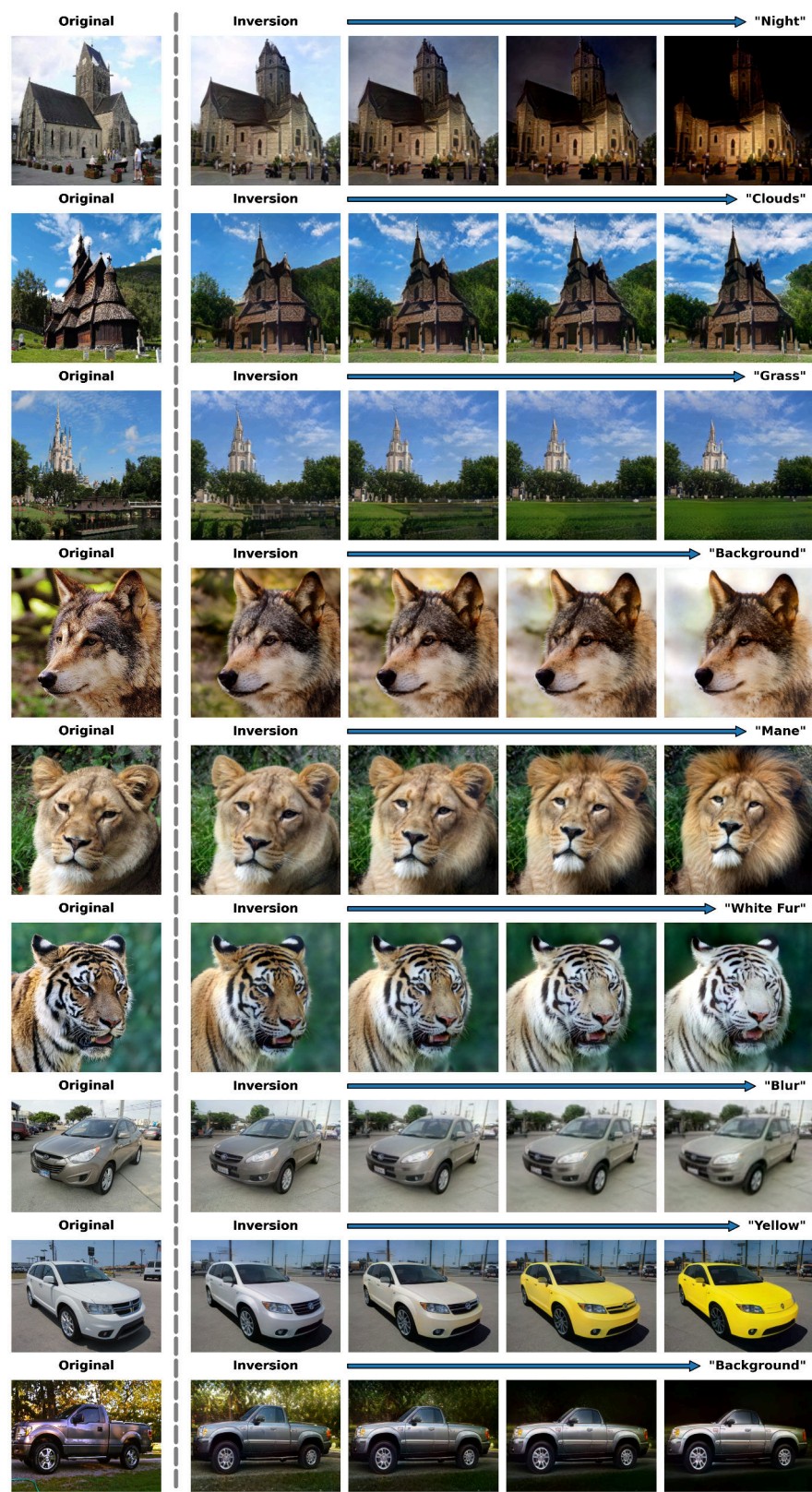

Figure 4: Qualitative interpolations on additional datasets. Rows show ICA directions applied to LSUN Church, Stanford Cars, and AFHQ Wild. Edits reveal coherent variations such as changes in lighting and environmental conditions in churches, body color and styling details in cars, and fur color, general attributes, and background color in animals.

## 5.5 QUALITATIVE RESULTS

To complement automatic analyses, we conducted a manual audit of all 512 ICA directions on CelebA by visually inspecting interpolation grids. A direction was marked *semantic* when it consistently induced a coherent, nameable change across multiple samples (e.g., eyeglasses, smile, hair). Roughly 30% of directions met this criterion, indicating that a non-trivial subset of high-contrast projections align with human-perceived attributes.

Interpolations along ICA directions reveal that the method captures many of the annotated ground-truth attributes in CelebA, while also uncovering additional semantic factors such as hair color (e.g., blond) or lighting conditions that are not part of the attribute labels (Figure 3).

Similar patterns are observed across other datasets: in LSUN Church, ICA uncovers directions that affect background and scene appearance; in Stanford Cars, directions modify object-level properties such as color and style; and in AFHQ Wild, directions capture animal traits as well as low-level image characteristics. Beyond these semantic factors, ICA also discovers directions that influence image-related artifacts, including sharpness or blurriness. Together, these qualitative results (Figure 4) illustrate that non-Gaussian projections consistently highlight meaningful and diverse aspects of variation, extending beyond labeled attributes and spanning both structural and appearance-related factors across domains.

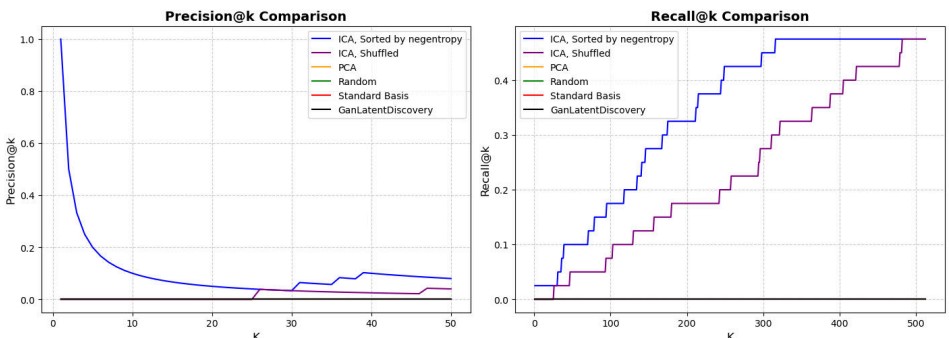

Figure 5: Precision@k (left) and Recall@k (right) for retrieving semantic directions. A retrieval is considered successful if at least one of the 40 hyperplanes learned from CelebA attributes has cosine similarity larger than 0.3 with the candidate direction. None of the baseline methods were able to recover semantic directions, whereas ICA consistently achieved non-trivial retrieval, with sorting by negentropy yielding the strongest results across all $k$.

## 6 CONCLUSION

The main question we wanted to answer is whether non-Gaussian projections of latent spaces contain semantic structure, and our findings suggest that they often do. Across CelebA, LSUN Church, Stanford Cars, and AFHQ Wild, directions selected for their departure from Gaussianity produced edits that were coherent and interpretable, ranging from facial attributes to lighting conditions, body color, and animal appearance. This outcome is somewhat surprising, given that the non-Gaussianity criterion is purely statistical and makes no explicit reference to image semantics.

At the same time, our study suffers from several limitations. The quantitative evidence was confined to CelebA, where attribute annotations provided ground-truth hyperplanes, while other datasets required purely qualitative evaluation. Moreover, not every non-Gaussian direction yielded a semantic edit, and the boundary between interpretable and noisy factors remains difficult to characterize.

Overall, these results suggest that statistical properties of latent codes can expose meaningful structure without supervision, pointing to an underexplored connection between classical tools such as ICA and modern generative models. Future work may investigate stronger theoretical links between non-Gaussianity and semantic alignment, explore alternative distributional criteria, and extend evaluation beyond faces and a handful of object categories to more diverse visual domains.

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

# A APPENDIX

## A.1 MANUAL AUDIT OF ICA DIRECTIONS

To complement automatic analyses, we conducted a manual audit of all 512 ICA directions on CelebA by visually inspecting interpolation grids. A direction was marked *semantic* when it consistently induced a coherent, nameable change across multiple samples (e.g., eyeglasses, smile, hair). Roughly 30% of directions met this criterion, indicating that a non-trivial subset of high-contrast projections align with human-perceived attributes.

We then examined whether ICA's intrinsic ordering by negentropy relates to the manual labels. Treating the set of human-annotated semantic directions as ground truth, we computed Precision@k and Recall@k while ranking directions by decreasing negentropy (Figure 8). Under our protocol, we did not observe a clear connection between higher negentropy and higher semanticity. Several factors could contribute: the manual pass may under-detect subtle but meaningful edits; our operational notion of "semantic" (privileging easily nameable attributes) may be too narrow; or ICA may surface smooth, low-salience variations that are systematic but harder to categorize.

Representative examples from the audit are shown in Figure 7, illustrating the range of effects captured by ICA. Overall, while this manual study suggests that ICA discovers many directions with coherent visual impact, our current evaluation does not establish a strong link between negentropy and human-labeled semanticity.

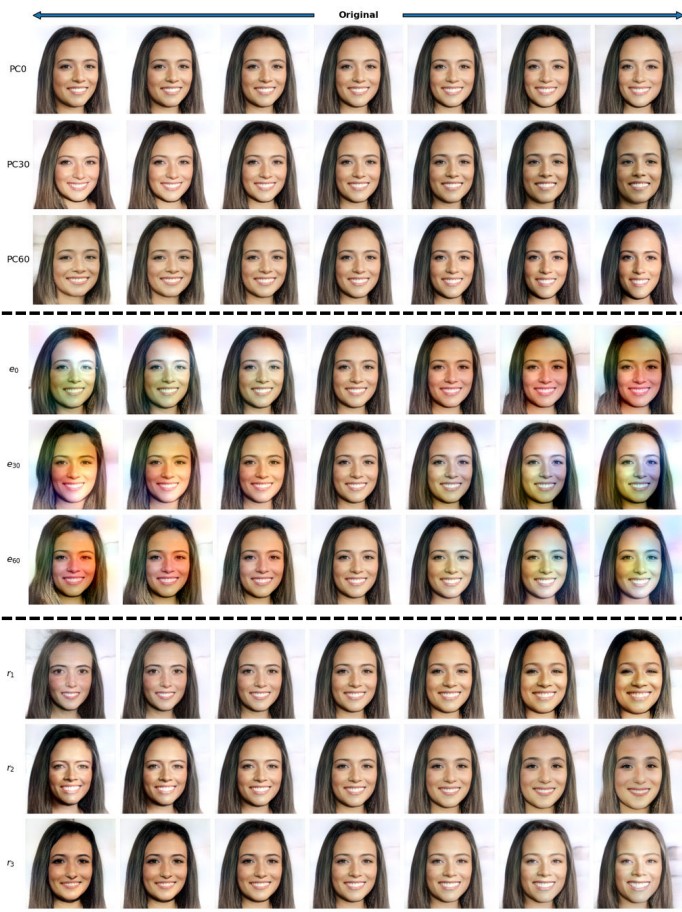

Figure 6: Rows 1–3 show interpolations of inverted CelebA samples along principal components in the latent space. Rows 4–6 show interpolations along standard basis directions, and Rows 7–9 along random directions.

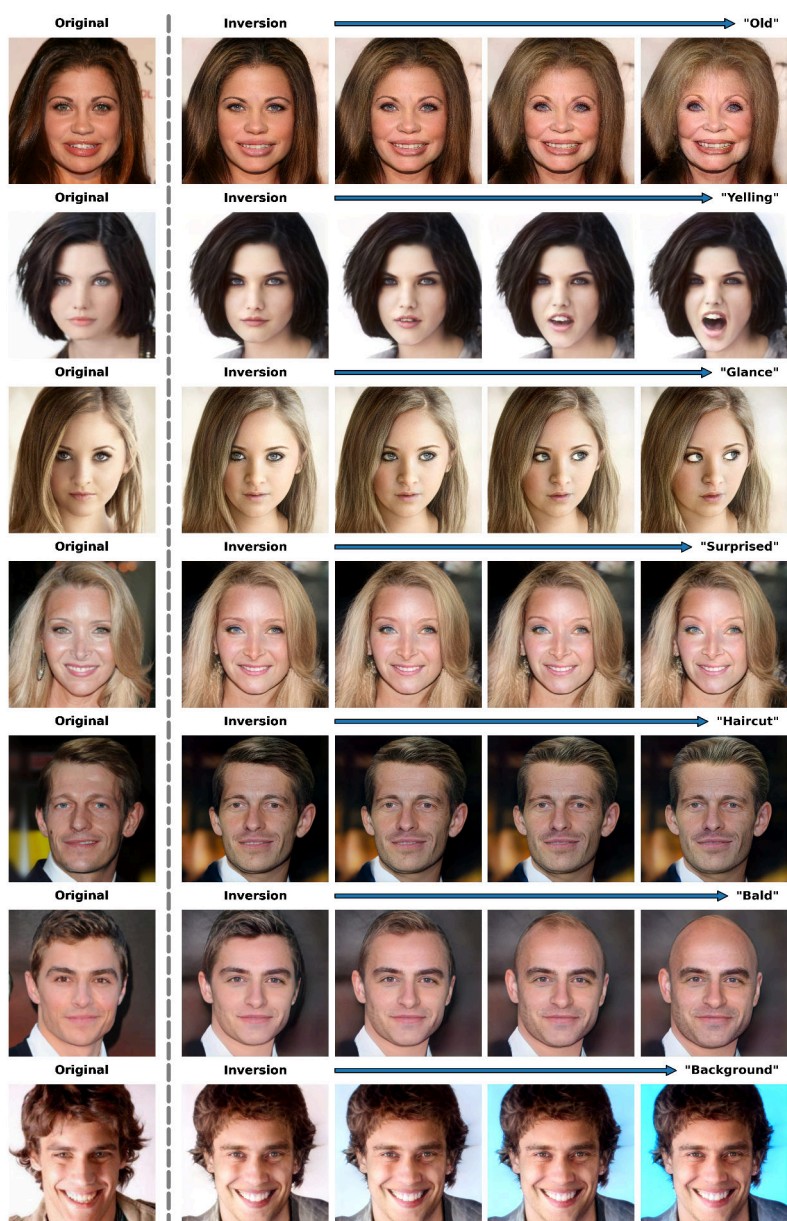

Figure 7: Additional qualitative results on CelebA. Each row shows a source image, its inversion, and interpolations along an ICA-discovered direction.

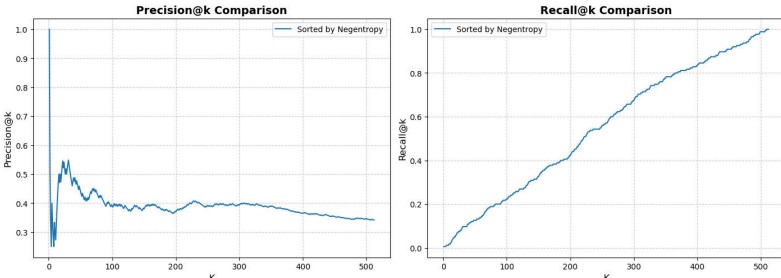

Figure 8: Additional qualitative results on CelebA. Each row shows a source image, its inversion, and interpolations along an ICA-discovered direction.

