# OpenReview forum: "Unsupervised Discovery of Visual Attributes Using Non-Gaussianity"
_ICLR.cc/2026/Conference — ICLR 2026 Conference Withdrawn Submission_

### Official Review · Reviewer_t3JA · 2025-10-23

**Soundness:** 2
**Presentation:** 2
**Contribution:** 1
**Rating:** 2
**Confidence:** 5

**Summary:**

This paper explores whether meaningful visual attributes, like "smiling" or "wearing glasses" can be discovered automatically in StyleGAN2 latent spaces. Building on the concept of projection pursuit, the authors search for linear directions in the StyleGAN2 latent space whose one-dimensional distributions are maximally non-Gaussian. Experiments across multiple datasets (faces, churches, cars, animals) show that these non-Gaussian directions often align with interpretable visual attributes, unlike raw latent coordinates or principal components.

**Strengths:**

N/A

**Weaknesses:**

Originality: Although the non-Gaussianity sounds new, the idea of exploring different latent directions for editing in StyleGAN2 has been well-studied and GANSpace [1] already achieved very similar results 5 years ago using PCA. Therefore, the originality of this work is low.

Significance: Image editing has made great progress in recent years, but this paper fails to recognize this and still focuses on an old model (StyleGAN2) using traditional methods (vector-based editing). The results are no better than those in the literature from five years ago [1]. Therefore, this research is of little significance.

Quality: The quality of the paper is low. GANSpace [1] used PCA and achieved good results, which has been reproduced by the community successfully. Why not compare with it? And why the PCA results shown in this paper are so poor?

Clarity: The clarity is okay but the content of this paper is very thin.


[1] Erik Harkonen, Aaron Hertzmann, Jaakko Lehtinen, and Sylvain Paris. Ganspace: Discovering interpretable GAN controls. NeurIPS 2020.

**Questions:**

Please see my comments above.

---

### Official Review · Reviewer_nXiH · 2025-10-31

**Soundness:** 1
**Presentation:** 3
**Contribution:** 2
**Rating:** 2
**Confidence:** 4

**Summary:**

The paper tries to find meaningful directions in the latent space of a pretrained StyleGAN2 model. The authors propose to search for directions that are as non-Gaussian as possible. As noted by the authors this idea in itself is not novel, but they provide evidence its useful for the StyleGan model in Figure 1, where directions derived from labelled data show non gaussian behavior (In fact they look like mixtures of two gaussians). To calculate the directions they make use of the standard fastICA algorithm. The experiments demonstrate that fastICA may in fact able to find useful directions in practice.

The paper is well written and the experiments answer most questions I had while reading the first part of the paper. However, because of the soundness of the experiments and the novelty of the paper, I can only recommend to reject the paper in its current form. I will detail my concerns in the weaknesses. If the authors address my concerns, I will be happy to raise my score.

**Strengths:**

- The paper is well written, it is easy to follow along and all steps seem to be well motivated. I only have a few questions as given below, but they are not central for the understanding of the paper.
- The proposed method is simple and builds upon standard and well research algorithms.
- The research questions addressed by the experiments are well thought out and I generally like the ideas behind them.

**Weaknesses:**

As indicated in the summary I have two major concerns, the novelty and the soundness of the experiments.

1. The novelty of the paper seems limited, not only because the idea itself is not novel, but also its application in modern settings. For example the cited work by Horan et al. does follows the same paradigm of Non-Gaussianity and also applies the fastICA algorithm. The authors say that in contrast to themselves, Horan et al. learn the directions from the data directly. However, not only is that a harder task as they claim themselves, but Horan et al. also use a manifold learning step before applying fastICA. In any case the distinction seems rather weak. Overall, this could still be interesting and I would like to hear the opinion of the other reviewers and the AC regarding the novelty.

2. Soundness of the experiments

   1. The biggest weakness of the paper are the qualitative results, and by that to some extent the motivation of the similarity threshold. As it is now we are only presented a very limited subset of the directions and for each direction only a single image. For a background direction there are several images, but its unclear if that is really the same direction. Ideally, the authors should release easily usable code that allows the reviewers to pick a direction and display a corresponding image grid for different base images. However, if this is really unfeasible I would also be happy to get at least four images for every direction in the supplemental material. Otherwise its just not possible to verify any of the claims made in the paper und crucially the failure cases of the proposed method. To better justify the threshold more data on the similarity threshold would be needed as well.

    2. The rather low precision score and the limited effect of the sorting by negentropy show that the measure might not be as strong as claimed.

   3. All non deterministic algorithms should be run several times, and appropriate error bars or areas need to be added to the plots.

   4. Ideally, the authors should release all code used in their experiments. In case they don't they need to at least specify software versions and algorithmic parameters in the appendix. The paper also does not explicitly specify the number of directions used for the baselines, this could be stated explicitly, but from context I assume it 512 for all of them.

   5. Something seems wrong with Figure 8, the data does not seem to fit the one from Figure 5. Apart from that the description is the same as the one for Figure 7.

Minor issues I noticed:
- Line 209 could be stated more clearly, I assume to compress the 18x512 to only 512, but the double use of dimension and distinct usage of style vectors for the 18 makes this less clear.
- To make the paper self-contained the definitions for Precision@k and Recall@k should be added to the appendix
- The description of Figure 3 should be improved, it is not entirely clear if you use the SVM directions for row 1-4 or if you matched your directions with the SVM directions and display your directions. I believe its the latter?

**Questions:**

- Why do you use logcosh? In [1] there are some arguments for why that might not be a good choice, but I should note that this is outside of my research area.
- How do you choose gamma in line 202?
- Did you try different levels of compressions with PCA? How does a further reduction influence the result?




[1] Wei, T. A study of the fixed points and spurious solutions of the deflation-based FastICA algorithm. Neural Comput & Applic 28, 13–24 (2017). https://doi.org/10.1007/s00521-015-2033-6

---

### Official Review · Reviewer_Vg85 · 2025-11-03

**Soundness:** 2
**Presentation:** 3
**Contribution:** 1
**Rating:** 2
**Confidence:** 4

**Summary:**

The paper investigates whether meaningful visual attributes can be automatically discovered from unlabeled image collections by leveraging projection pursuit. They search for linear directions in the latent space of a pretrained StyleGAN2 whose one-dimensional projections are maximally non-Gaussian, hypothesizing that these directions correspond to semantic concepts. Experiments across multiple datasets show that non-Gaussian directions often align with interpretable attributes, unlike raw latent coordinates or principal components.

**Strengths:**

* The writing is clear, well-organized, and easy to follow throughout the paper.
* The paper is well-motivated with supporting quantitative and qualitative experiments.
* The authors conduct analyses across datasets spanning multiple domains to support the generality of the method.

**Weaknesses:**

* The paper offers limited novel contributions. It primarily applies existing methods without introducing substantial conceptual or technical innovations. Furthermore, the analysis focuses on a single model, StyleGAN2, which is relatively outdated and no longer widely adopted in the vision community. Its contributions and empirical findings lack the level of novelty and impact for acceptance at ICLR.

* The paper is missing baseline comparisons, such as Bahng et al. [1], Bau et al. [2], which also explore unsupervised discovery of visual attributes. It remains unclear how the proposed approach improves upon or differs from such existing methods.

* It is unclear how meaningful the SVM weight vectors are as ground-truth semantic directions, and the interpretability of the discovered attributes is not well supported. As an alternative, the authors could perform attribute classification using pre-trained models, as demonstrated in [1] (Section 3.5). In modern settings, they could leverage state-of-the-art vision–language models (e.g., Qwen3-VL) via prompting or train a linear classifier on top of strong visual encoders such as DINOv3.

* There are several typos (e.g., lines 014, 032, 052).

[1] Exploring Unlabeled Faces for Novel Attribute Discovery, CVPR 2020.

[2] GAN DISSECTION: VISUALIZING AND UNDERSTANDING GENERATIVE ADVERSARIAL NETWORKS, ICLR 2019.

**Questions:**

Addressed in the weakness section

---

### Official Review · Reviewer_ZfGD · 2025-11-05

**Soundness:** 2
**Presentation:** 3
**Contribution:** 2
**Rating:** 2
**Confidence:** 4

**Summary:**

This paper proposes to use non-Gaussianity as a way to identify visually meaningful attributes by projecting StyleGAN2 latent codes to such linear directions. The overall idea follows the classical projection pursuit and the paper shows the method is capable for unsupervised discovery of visual attributes.

**Strengths:**

The method seems to work reasonably well and shows improved performance compared with baselines.

**Weaknesses:**

The method is only validated using StyleGAN2. It is unclear whether the conclusions are generally applicable to state-of-the-art models (GAN, diffusion based, etc.)

Out of the top 500 directions in terms of non-Gaussianity, only 30% has clear meanings. There are also no evaluation to show that such directions cover most visual attributes, so the practical use for the method is unclear.

The major algorithm is the existing FastICA. With the questionable potential use, the technical contributions of the paper are limited.

**Questions:**

Why is this method useful in practice, especially compared with methods such as

CLIP2StyleGAN: Unsupervised Extraction of StyleGAN Edit Directions

---

### Note · Authors · 2025-11-13

I have read and agree with the venue's withdrawal policy on behalf of myself and my co-authors.